# Long-Term Device Satisfaction and Safety after Cochlear Implantation in Children

**DOI:** 10.3390/jpm12081326

**Published:** 2022-08-18

**Authors:** Milan Urík, Soňa Šikolová, Dagmar Hošnová, Vít Kruntorád, Michal Bartoš, Petr Jabandžiev

**Affiliations:** 1Department of Pediatric Otorhinolaryngology, University Hospital Brno, Černopolní 9, 61300 Brno, Czech Republic; 2Faculty of Medicine, Masaryk University Brno, Kamenice 5, 62500 Brno, Czech Republic; 3Department of Pediatrics, University Hospital Brno, 61300 Brno, Czech Republic

**Keywords:** cochlear implants, device use, pediatric, audio processor satisfaction questionnaire (APSQ), cumulative survival

## Abstract

(1) Objectives: For full benefit in children implanted with a cochlear implant (CI), wearing the device all waking hours is necessary. This study focuses on the relationship between daily use and audiological outcomes, with the hypothesis that frequent daily device use coincides with high device satisfaction resulting in better functional gain (FG). Confounding factors such as implantation age, device experience and type of device were considered. (2) Results: Thirty-eight CI children (65 ears) were investigated. In total, 76.92% of the children were using their device for >12 h per day (h/d), 18.46% for 9–12 h/d, the remaining for 6–9 h/d and one subject reported 3 h/d. The revision rate up to the 90-month follow-up (F/U) was 4.6%. The mean FG was 59.00 ± 7.67 dB. The Audio Processor Satisfaction Questionnaire (APSQ) separated for single unit (SU) versus behind the ear (BTE) devices showed significantly better results for the latter in terms of wearing comfort (WC) (*p* = 0.00062). A correlation between device use and FG was found with a device experience of <2 years (*n* = 29; r^2^ = 0.398), whereas no correlation was seen with ≥2 years of device experience (*n* = 36; r^2^ = 0.0038). (3) Conclusion: This study found significant relationships between daily device use and FG, wearing comfort and long-term safety (90 months).

## 1. Introduction

Cochlear implants (CIs) have been used for more than 40 years. Nowadays, state-of-the-art technology allows CI recipients to achieve high levels of speech understanding, enables normal social interactions, and enhances cognitive and linguistic development, which in turn significantly improves quality of life in CI recipients [1,2]. It is well accepted that in the pediatric population, early and consistent stimulation of the cochlea is critical for the optimal development of speech and language [3,4]. Hearing loss (HL), especially in young children, poses a barrier to education and social integration, as it affects language acquisition, all of which impact total score literacy and the child’s self-esteem and social skills [5,6,7]. Untreated hearing loss is often associated with academic underachievement, which can lead to reduced employment opportunities later in life [8,9,10]. It is widely accepted that early identification of hearing loss in children, e.g., through hearing screening programs, followed by timely and appropriate interventions can minimize developmental delays and facilitate communication, education, and social development [4,5]. Research suggests that children who are born deaf or acquire hearing loss very early in life and who receive appropriate interventions within six months of age are at par with their hearing peers in terms of language development by the time they reach the age of five years (in the absence of other impairments) [4,5,11,12,13]. However, surgery and CI adaptation alone do not guarantee the full benefit of its users. Research has shown that several factors may interfere with the performance and quality of life of implanted children. Almost one third of the children implanted with a CI suffer from issues unrelated to hearing loss that affect their auditory and cognitive development. Papsin et al. reported that in the literature, 20% of children with sensorineural hearing loss have had associated radiological anomalies of the temporal bone. These temporal bone anomalies are associated with a wide range of hearing acuity, varying degrees of progression of hearing loss, and presence or absence of related non-otological anomalies [14]. Besides the widely discussed etiology-related reasons for various degrees of beneficial performance of implanted children, also age at surgery, time of CI activation, time of auditory deprivation, access to speech therapy as well as family engagement in the therapeutic processes, and the time of daily use of the device (hence, the real benefit to its user) have been highly discussed as confounding factors [15,16,17]. The here investigated study cohort, with a rather high mean age at implantation of 6.60 ± 3.40 years (range: 1–18), underlines once more the many different impairment factors when it comes to the beneficial development of hearing, here apparently caused by long wearing times and user satisfaction. The influencing factors remain to be verified. Previous subjective questionnaires such as the Speech, Spatial and Qualities of Hearing Scale (SSQ), the Hearing Implant Sound Quality Index (HISQUI19), or the LittlEARS Auditory questionnaire (LEAQ^®^) do evaluate the subjective quality of hearing but do not actually reflect the users’ satisfaction, which might in turn possibly increase the motivation to wear the device for longer [18,19,20,21,22,23,24]. Recently, Obrycka et al., 2022, reported on Audio Processor Satisfaction pre and post AO upgrade and found no significant difference between the device generations [25]. The here presented study therefore aimed to investigate the pediatric users’ satisfaction with their audio processor (AP) and to correlate with several possibly influencing factors such as wearing time, age at implantation, and follow-up period. 

## 2. Materials and Methods

### 2.1. Study Population

The retrospective data analysis following implantation was performed as part of routine clinical procedures between 2018 and 2021 at the university clinic. The audiological inclusion criteria were based on manufacturer’s recommendations; pediatric patients with severe-to-profound hearing loss were included (Figure 1). The study protocol to administer the APSQ was approved by the ethics committee of University Hospital (No. 12-130218/EK) and informed consent of the parents/legal guardian was given.

### 2.2. Audiological Evaluations

All audiometric tests were performed pre-operatively (pre-OP) and post-operatively (post-OP) in a soundproof audiometric booth, using the audiometer Interacoustics AC40E (Denmark, 2019). Pure tone measurements were performed at a frequency range from 0.5 to 4 kHz. Pure tone average air conduction hearing thresholds (PTA4AC) were calculated as the mean of the evaluated AC values at 0.5, 1, 2 and 4 kHz. Soundfield thresholds were measured using frequency—modulate warble tones presented from the aided side, with the loudspeaker positioned 3 m away from the subject. Soundfield audiometry (SF) at 65 dB HL in a multi-talker babble was performed. Pre- and post-operative SF outcomes were used to calculate functional gain (FG).

### 2.3. Satisfaction Related Questionnaire

The APSQ comprises 15 items, which were classified according to their content into the subscales: comfort, social life, and usability, and the resulting total score dimension. Each subscale contains five items. The subscale scores and the total score are obtained by calculating the average scores provided on each item. A visual analogue scale (VAS) between 0 and 10 for better users’ satisfaction differentiation with their AP. A score of 0 corresponded to a response of ‘does not agree at all’ and 10 as ‘fully agrees’. If an item did not apply to the subject, then the subject could tick the ‘not applicable’ option or leave the scale blank. The questionnaire was completed at the clinic by the teenage users themselves or by their parents for younger children [26].

### 2.4. Data Analysis

Descriptive analysis was used to report demographics (e.g., age and gender), baseline characteristics (e.g., aetiology), and patient-reported outcome mean, standard deviation (SD), median, minimum, and maximum (Table 1). The non-parametrically distributed outcomes were analyzed using GraphPad Prism 7.0 statistical software (GraphPad Software, San Diego, CA, USA, www.graphpad.com). The Wilcoxon signed-rank test was applied to evaluate significant differences between unaided (pre-OP), and CI-aided (post-OP) sound-field outcomes (Table 2, Figure 1). Scores from the APSQ were analyzed using the Mann–Whitney test to test for significant difference (Table 2). For the results on FG depended on follow-up (F/U) time linear regression analysis was performed to correlate measures of daily wearing time and functional gain and device experience (<2 years and ≥2 years) (Figure 2).

The APSQ questionnaire outcomes and device used (single unit (SU) versus behind the ear (BTE)) are displayed in Figure 3.

APSQ outcomes for each dimension (wearing comfort, social life, usability and total score) are displayed in boxplots, with the ends of the box representing the upper and lower quartiles (interquartile range), the vertical line inside the box marks the median and the whiskers extend from the highest to the lowest observationan the individual outcomes are displayed as circles within the boxplot (Table 3, Figure 4).

For the safety outcomes, a Kaplan–Meier Survival analysis was performed calculating for each time interval a ‘survival’ probability calculated as the number of subjects surviving divided by the number of patients at risk. Survival rates were separated into implant survival and external parts survival and displayed separately within the same graph (Figure 5).

## 3. Results

### 3.1. Subjects and Surgical Results

Pre-operative and follow-up data were fully available for 38 subjects, out of which 27 were bilaterally implanted (65 ears in total). The mean age at implantation was 6.60 ± 3.40 years (range: 1–18), and the group comprised 20 males and 18 females.

The mean hearing age of the investigated study group was 2.40 ± 1.78 (range: 0.5–7). Audiological outcomes for two different follow-ups were available with the first mean F/U being at 1.11 ± 0.31 months (range: 0.5–1.5) and the second mean F/U at 2.23 ± 1.62 months (range: 0.5–7.0). The APSQ was administered at 2.33 ± 1.78 years post op. Complications were recorded up to 90 months (7.5 years) after surgery. A total of 31 subjects reported congenital HL, out of which 22 genetic causes with family background were identified. For the remaining 7 subjects, cause and onset of HL is unclear (idiopathic), and the subjects were implanted between 4 and 18 years of age (mean 9.30 ± 4.76). The causes of the severe-to-profound hearing loss of the study population could be attributed to the Mondini syndrome in two children, acquired hearing loss was reported in five cases, the reasons being meningitis (*n* = 1), premature birth (*n* = 2), cytomegalovirus infection (*n* = 1), and common cavity (*n* = 1). The AP distribution of the study cohort was eight Opus 2 users, 14 Sonnet and 38 Rondo 2 users, and five RONDO 3 users. Out of the 38 investigated subjects, 77% reported a use of >12 h/day, 18% reported a use of between 9 and 12 h/day, and 3% a use of between 6 and 9 h/day. One user (2%; Mondini case) reported to wear his AP for 3 h per day. The high daily use is an indicator for the satisfaction and was also displayed by the “Wearing Comfort” outcomes examined via the APSQ questionnaire bythe users. No complications occurred during surgery.

### 3.2. Audiological Outcomes

Before implantation, the study cohort reported a severe hearing loss of 90.13 ± 3.08 dBHL. The preoperative PTA4 Soundfield improved at the first F/U (1.30 ± 0.24 years) to 35.75 ± 7.03 dBHL and to 31.13 ± 7.30 dBHL at the second F/U (mean 2.18 ± 1.31 years) (Table 1, Figure 1).

#### 3.2.1. Functional Gain in Relation to Age at Implantation

This results in a mean functional gain of 54.38 ± 6.91 dB at the first F/U and of 59.0 ± 7.67 dB (Table 1) at the second F/U. The mean FG at PTA4 was not significant but, at the frequencies 0.5, 2 and 4 kHz, significant differences between the age at implantation below (<3 years) and above (≥3 years) the age of three in favor of the latter at the last F/U were observed (*p* < 0.05). At 5 kHz, the improvement in FG significantly improved even within the older implanted population (F/U 1.30 vs. 2.18 years; *p* < 0.05).

#### 3.2.2. Functional Gain in Relation to Length of Follow-Up

Investigating the FG over time, a significant improvement can be seen between the F/U in the first year compared to the F/U in the third and fourth year (*p* = 0.014). A further significant improvement in FG over time was seen at the time point 6 to 7 years post implantation compared to the first year (*p* = 0.0126).

#### 3.2.3. Functional Gain in Relation to Daily Wearing Time

Investigating the correlation between FG and daily wearing time of the device showed a slight correlation of r^2^ of 0.3983 when subjects with CI experience under two years were investigated. After using the device for two or more years, both the FG and the wearing time seems to reach a plateau with no correlation r^2^ = 0.00385 (Figure 2). Emphasis needs to be placed on the fact that only in the group of users with an experience of <2 years the wearing time was below 10 h per day and the lowest FG of 33.75 dB was observed. For the group of experienced users (≥2 years) wearing time was more than 12 h per day.

### 3.3. Audio Processor Satisfaction Questionnaire (APSQ)

The APSQ was administered approx. 2.33 ± 1.78 years after implantation (range: 6 months–7 years) (Table 2, Figure 4). The 15 questions are separated into the dimensions of “Wearing Comfort”, “Social Life”, “Usability”, and the “Total Score”.

### 3.4. APSQ as a Function of Type of Audio Processor

Outcomes were separated into single unit (SU) users (Rondo 2; *n* = 43) versus behind the ear users (BTE) (Sonnet, OPUS 2; *n* = 22). The “Wearing Comfort” was significantly better with the BTE device (*p* = 0.0062), whereas the other dimensions were not significantly different from each other (Figure 3).

### 3.5. APSQ in Relation to Length of Follow-Up

Investigating the “Wearing Comfort” over the total F/U of 7 years, a significant improvement from the first year to the 3rd and 4th year was observed (*p* = 0.0053). The same was seen comparing F/U year 1 and 2 to the 3rd and 4th year (*p* = 0.053).

Social Life significantly improved from year 1 and 2 to the 3rd and 4th year (*p* = 0.0267). For the dimensions of usability and the total score outcomes, scores were from the beginning very high, with a mean total score of 9.98 ± 3.4 and 9.21 ± 1.35 from a maximum of ten, respectively, and no significant differences over time were seen (Figure 4).

No correlation was found when investigating daily use versus age at implantation or age at the time-point of questionnaire administration (r^2^ = 0.00131, r^2^ = 0.00167).

### 3.6. Safety

For the safety analysis, the outcomes were split between (1) major complications requiring surgical attention, (2) minor complications with a medical background such as reoccurring otitis media and (3) external device-related complications such as too strong a magnet (Table 3) (Figure 5).

*(1)* 
*Major complications*


Out of the 65 implanted ears, three major complications requiring surgical attention occurred (4.6%). The revisions were performed within less than one month after occurrence and no further problems were reported. Explantation without reimplantation did not occur in any of the analyzed cases.

*(2)* 
*Minor complications: medical background related*


Minor complications regarding the external device compartment, which were resolved by reducing the magnet strength, were reported in five cases (7.70%). In seven cases, an underlying medical problem such as otitis media, retroauricular phlegmon, and steamy skin was reported, all of which were successfully treated with local disinfection and/or antibiotics (10.77%).

All minor complications were solved within a mean of 4.60 ± 6.04 days. The ‘survival’ probability calculated via Kaplan–Meier Survival analysis is shown in Figure 5, and all complications, also including the external device breakage, are shown.

*(3)* 
*Minor complications: external device related*


The survival proportion of external parts/minor complications (incl. change of magnets) was up to 19 months at 90.00%. For the remaining 71 months, up to a total of 90 months, the external device survival proportion was 86.65%. Investigating the minor events with an underlying medical condition resulted in a survival proportion of 78.77% at 90 months F/U. Up to 31 months post-operative, this proportion was at 91.66%.

The outcomes clearly reflect the safety of the procedure in the pediatric population in terms of major complications with a cumulative survival rate of 90% at 85 months. The cumulative survival was stable with 96.96% up to 40 months when a dysfunctional electrode was reported. The first revision occurred 6 days after surgery due to wound dehiscence and the necessity to re-suture the wound. The third and last complication occurred 40 months post implantation due to a head injury leading to an electrode short-circuit. Over a course of 39 months, the survival proportion of the major complications was 96.92%. This section may be divided by subheadings. It should provide a concise and precise description of the experimental results, their interpretation, as well as the experimental conclusions that can be drawn.

## 4. Discussion

Numerous studies have identified a link between daily device use and communication outcomes in pediatric CI users [27]. Several studies in pediatric CI users reported the positive correlation of daily device use with speech recognition skills, and expressive and receptive language [28,29,30,31]. As a further consequence, one could also assume that frequent device use goes hand in hand with high device satisfaction. Currently, most studies concerning pediatric CI users have focused on assessing auditory skills and speech including verbal language ability or communication skills but never linked outcomes with quantitative device satisfaction and possibly related device use as well as device type dependency [4,8,13,27,31,32,33,34].

Until now, satisfaction measures for APs have only been designed for specific products, such as the RONDO device-specific questionnaire, but not across different APs and hearing systems or across different product generations [35,36]. Only the recent study by Obrycka et al. compared mean APSQ post-upgrade to pre-upgrade scores and showed an increase by 0.12 points for total score, 0.09 for comfort, 0.16 for social life, and 0.11 for usability. The increase in the social life dimension was found to be significant [25]. The authors also reported a higher level of functioning in different everyday lifestyles, less hearing disability, and more satisfaction with new audio processors, particularly in social situations. Interestingly our study cohort, comprising six different AP generations, two SU and three BTE devices, showed no differences in objective measures such as audiological outcomes or daily wearing time, but subjective impressions regarding wearing comfort were significantly different. Apparently, the BTE APs, namely, the Sonnet, the Opus and Opus 2, were rated with significantly higher wearing comfort when compared to the SU APs, the RONDO (both generations).

Due to the low age of the subjects, no speech audiometry was performed; instead, the FG was calculated as a measure of hearing benefit. The measured mean FG of 59.00 ± 7.67 dB (ranging from 33.75 to 80) is on the upper level of the reported outcomes in the literature underlining the beneficial impact of the procedure. Age at implantation shows conflicting findings across studies, with positive and negative correlations between age at implantation and daily device use [27,31]. Niparko et al. reported a positive association of spoken language with earlier, as opposed to later, age at implantation and greater residual hearing prior to implantation, but also showed that associations with environmental factors were evident as well [37]. Wiseman, on the other hand, reported a strong relationship between daily device use and early communication. The authors suggested that intervention strategies must also consider barriers to consistent device use and insisted on support for young children with CIs who struggle with inconsistent device use [27]. Karltorp et al. investigated children implanted between 5 and 11 months of age and showed an age-equivalent level of language understanding and improved vocabulary outcomes, which was earlier than when compared to children implanted between 12 and 29 months [38]. Nicholas et al. also analyzed vocabulary, expressive and receptive language at 4.5 years of age and showed that children implanted between 6 and 11 months (*n* = 27) achieved higher scores on all measures compared to those implanted between 12 and 18 months (*n* = 42) [39]. Numerous studies demonstrated similar results regarding improved language trajectories among infants implanted <12 months [27,38,40,41,42,43]. However, Leigh et al. showed no significant difference in the rate of receptive language growth between children implanted <12 months compared to those implanted between 13 and 24 months [44]. These results were similar to previous studies suggesting a limited advantage of performing cochlear implantation <12 months compared to those implanted between 13 and 24 months [45,46]. Wie et al. showed that in the first 4 years after implantation, the language performance of children with CIs was similar to that of their NH peers. However, between 4 and 6 years after implantation, the authors reported certain challenges with aspects of language, specifically receptive vocabulary and expressive grammar [31]. The limitation of some of these results is probably the direct comparisons between groups that reflect differences in the length of device experience: early implanted subjects have more time to develop language compared to infants implanted at a later age when assessment time (F/U appointment) is the same for both [47]. The great variability observed among the implanted children is probably related to the differences in age of the intervention, type of device, time and model of language therapy received after receiving that implant.

The literature also suggests that while early implanted children show a similar or even better language than their NH peers at the beginning of their rehabilitation, with time this reaches a certain peak, which is then also met by late implanted children as they tend to progressively compensate for the delay in the years following implantation [48,49]. Albegger et al. even compared, in a long-term longitudinal study, the impact of cochlear implantation on the educational placement, vocational outcomes and employment status of hearing-impaired adolescents and young adults. The investigated mean study population age was comparable to the age group investigated herein, ranging between 5.2 and 8.5 years at time of implantation. The authors reported that the majority of up to 81% attended a regular main-stream school and 27% even attended university, and concluded that the educational level of even late implanted Cl users does not differ from the NH population [50]. Our long-term outcomes over 90 months are in agreement with the above-mentioned and showed a correlation between daily device use in hours and FG as a factor of device experience and not implantation age (<2 years), whereas no correlation and a sort of plateau was reached for those with a device experience of ≥2 years. The literature suggests that significantly poorer device use is accompanied with demographic factors such as younger chronologic age, lower maternal education, presence of additional disability, and even sign-focused communication mode, etc. [27,31,51]. The authors investigated factors in addition to older age at implantation, such as satisfaction with the external component in analyzing the BTE and SU processors. While the majority (almost 70%), especially since 2019, of our CI children wear SU processors such as the RONDO (all generations) (mean 11.51 h/day versus 11.29 h/day for BTE devices), no clear correlation was found between the AP and reduced or increased wearing time. Even though the reported daily device use was high, information on fatigue involved in ‘electrically’ processing sound was not evaluated but should receive more attention form medical specialists and speech therapists, especially during language development. One concern, especially for the investigated age group of users, was the safety in terms of the holding strength of the SU compared to a slightly more stable wearing option through the additional ear hook fixation. This hypothesis could not be confirmed, as in the RONDO users only one AP needed to be replaced due to falling to the ground and one in case of an Opus device (BTE system). Noteworthy is the fact that out of the five reported AP magnet changes required, except for one, all were wearing the SU-System, the RONDO. Based on the correspondence with the families, the magnet strength chosen was too strong due to the fear of losing the device. With the first appearance of redness or uncomfortable feeling around the magnet site, the AP-magnet was changed without the occurrence of pain, scalp atrophy, or skin breakdown. As already mentioned above, the administered APSQ evaluating the “Wearing Comfort”, Social Life, and usability was therefore split between the BTE and SU users. Only the “Wearing Comfort”, a group of questions which might also reflect the safety feeling of aforementioned BTE component, showed a significantly better outcome in the BTE group compared to the SU RONDO users. Again, emphasis needs to be placed on the heterogenous distribution of study groups (*n* = 22 vs. *n* = 43; respectively). This imbalance between the devices also needs to be taken into account when looking at the complications concerning external components. At first glimpse, it appears as if there might be a correlation with the SU devices and external problems, but this might be solely due to the unequal distribution. Parents, when choosing an AP for their children, probably prefer the RONDO device (all generations) for cosmetic, non-stigmatizing reasons and because the wearing of one unit might be less bothersome for small children. Our results clearly show the preference in terms of “Wearing Comfort” measured via the APSQ. Interestingly, neither “Social Life” nor “Usability” were significantly different, hence also not the “Total Score” when separating users based on their device (BTE vs. SU). When investigating the time of use, however, significant differences can be seen in “Wearing Comfort”, with improvement from the first and the second year to the fourth and fifth year. Moreover, the score of “Social Life” significantly improved from the second to the fourth year, which might be due to the users’ age and their more social involvement after 4 years. A number of studies reported that children with a CI were able to attend mainstream education [52,53], a fact that we are planning to investigate with the here discussed study cohort in another independent future study. Another important point should be noted that not only rehabilitation, but also audio processor (AP) fitting is a process which takes time and requires patience and determination, not only from the patient themselves but also from the parents as well as their peers. The authors also collected long-term safety information and separated complications into major and minor medical complications (usually taking longer to heal) versus external device breakage, which, depending on the availability of spare parts/spare devices, may be fixed immediately. While the mean time of a minor complication was 4.6 ± 6.04 days (between 0, equivalent to instant solution up to 20 days, where a special magnet strength needed to be ordered), the time required to solve major complications was 42.33 ± 54.96 days on average (between 6 and 120 days, latter due to increased waiting time for a surgery appointment because of the COVID situation). Since the magnet problem was, except for one Sonnet AP case (BTE), only reported for the RONDO2 AP (SU), it stands to reason that for safety purposes and the fear of losing the AP, a too-strong magnet was selected by the parents. The incidence of complications is inversely related to the age of the pediatric patient, which is in our study cohort only related to the external device complications. Major complications requiring revision surgery occurred in 3 out of 65 ears, which translates to a rate of 4.6%. While this appears relatively high, it is comparable to literature reports and somewhat explainable looking into detail of the cause of complications [54,55]. For example, a head injury leading to electrode short-circuit or electrode dysfunction and the third case of wound dehiscence 2 days after surgery might not be directly related to the device rather than to poor wound healing. Nonetheless, cochlear implantation is a safe surgical technique for the rehabilitation of severe-to-profound sensorineural hearing loss in children. The global minor complication rate was reported with 14.9 and 5% for major complications.

## 5. Conclusions

This study found significant relationships between daily device use (measured via questionnaire) and three separate outcomes: (1) early auditory skills measured as FG, (2) wearing comfort, social interaction, and usability elicited from the APSQ and (3) long-term safety measured as cumulative survival up to 90 months in young children with a CI. Confounding factors such as implantation age, device experience, and type of device (SU versus BTE) were considered. Finally, this study identified possible factors related to daily device use. This article provides additional long-term evidence for the importance of consistent device use in young CI users and strengthens the clinical profile regarding the safety of the intervention: no intra-operative complications occurred, and the rate of major complications was very low.

## Figures and Tables

**Figure 1 jpm-12-01326-f001:**
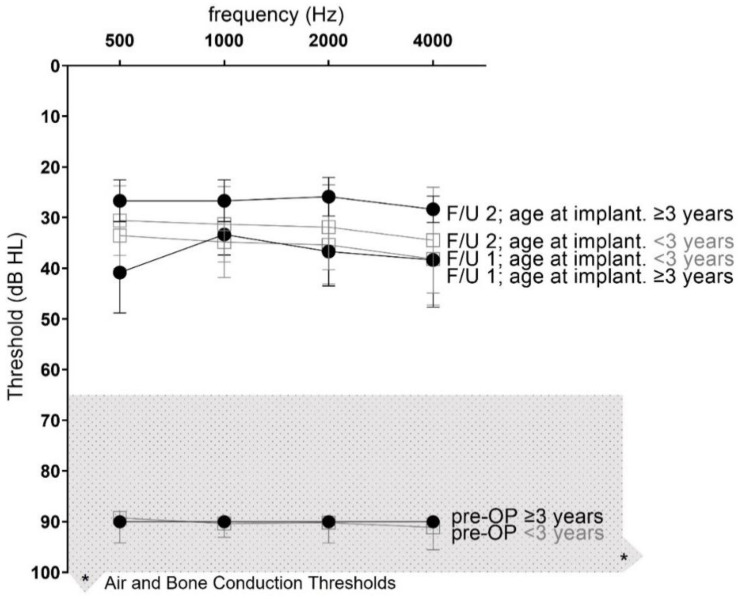
**Soundfield audiometry.** Displayed is the Soundfield audiometry showing the mean pre- and post-operative outcomes for two different follow-up (F/U) times: F/U1 (black dot): mean 1.30 ± 0.24 years; F/U2 (open square): 2.18 ± 1.31 years; separated regarding the age at implantation; full sign: age at implantation 3 years or older (≥3); open sign: age at implantation under 3 years of age (<3). The groups respective pre-operative values (unfilled symbols) are presented to show that subjects were within. The manufacturer’s indication limits are given as a light gray box, upper and lower indication limit in dB HL and * indicates upper limits in the *x* and *y* coordinates of up to 120 dB and up to 8 kHz.

**Figure 2 jpm-12-01326-f002:**
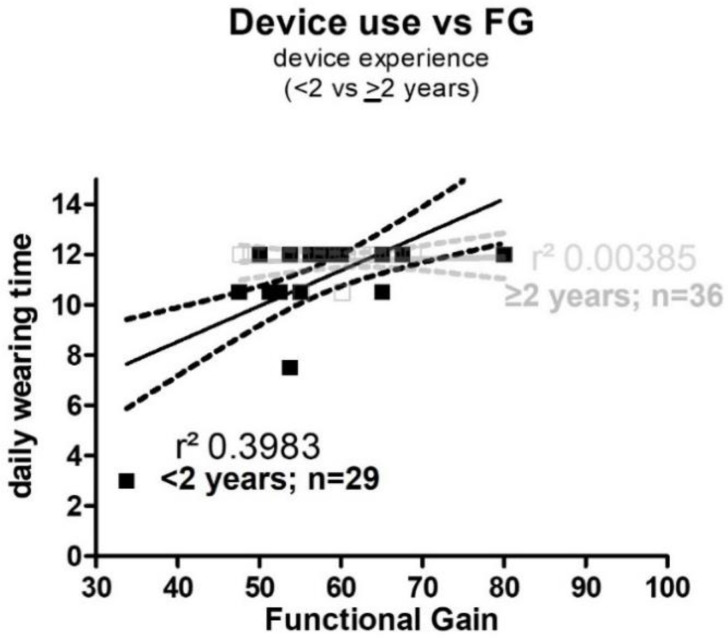
**Scatter plot correlation functional gain (FG) vs. daily device use.** The scatter plots and the corresponding linear regression (full line) and its upper and lower confidence interval (dotted line) shows the correlation between the mean functional gain (FG) average and the daily device use in hours. In dark squares are the individuals with a device experience of less than two years (<2) and the gray open squares represent the children with a device experience of 2 years or more (≥2). The correlation coefficient r^2^ was 0.3983 for the group <2 years (slight correlation) versus and r^2^ of 0.00385 for the group of ≥2 years (no correlation).

**Figure 3 jpm-12-01326-f003:**
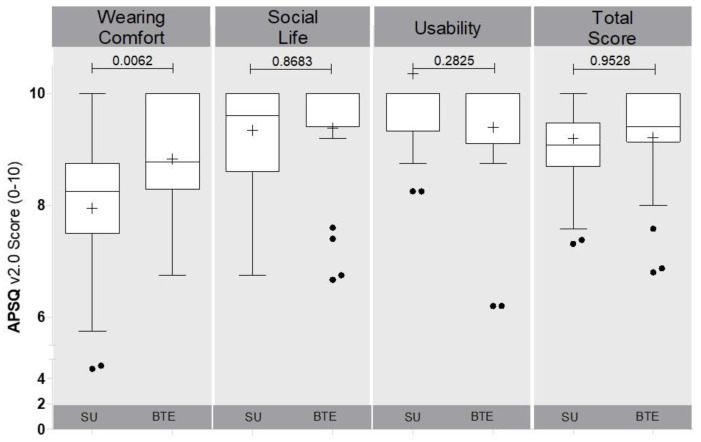
**Audio Processor Satisfaction Questionnaire (APSQ) as a function of device used.** Boxplot shows the distribution of outcomes from the Audio Processor Satisfaction Questionnaire (APSQ) separated for the type of device used (SU (Single Unit) vs. BTE (Behind the Ear)) and split for the dimensions of “Wearing Comfort”, “Social Life”, “Usability” and the resulting “Total Score”.

**Figure 4 jpm-12-01326-f004:**
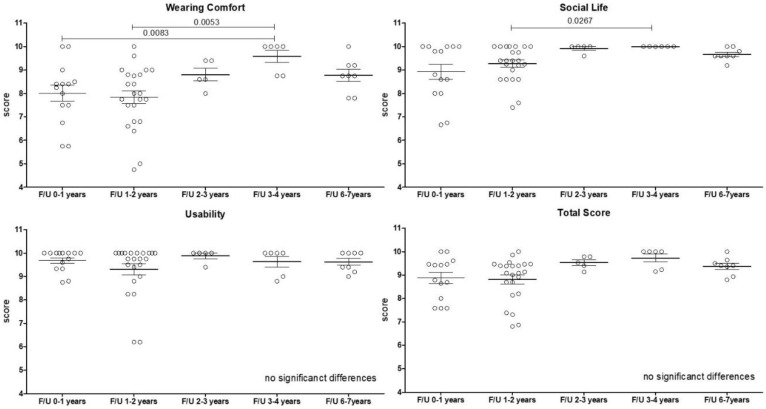
**Audio Processor Satisfaction Questionnaire (APSQ) as a function of follow-up (F/U).** Scatter boxplot showing the dimension outcomes of the Audio Processor Satisfaction Questionnaire (APSQ) over time. F/U (Follow-up).

**Figure 5 jpm-12-01326-f005:**
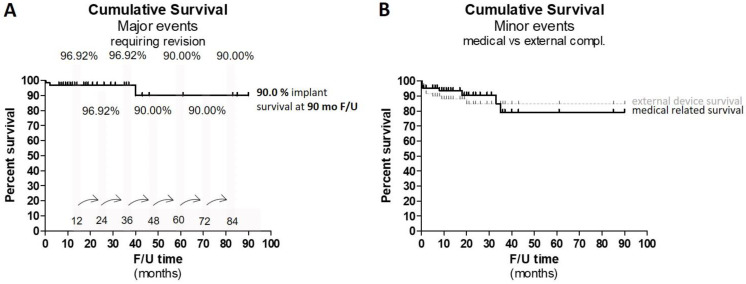
**Kaplan–Meier survival analysis.** Graph shows the Kaplan–Meier survival analysis calculating the ‘survival’ probability (**A**) for major complications over a course of 90 months. The survival proportion was extracted for each event shown in great bars and indicated by the arrow at the respective event time (months). Up until 36 months, the survival proportion lies at 96.92% and drops then for the remaining 54 months (total 90 months) to 90.0%. (**B**) The ‘survival’ probability for minor complications, split into medical-related conditions and external device problems (too strong a magnet). The survival proportion of the external device survival lies at 86.648%, whereas the survival proportion for the medical-related condition lies at 78.769%.

**Table 1 jpm-12-01326-t001:** Soundfield audiometry (PTA 4).

		Follow-Up 1			Follow-Up 2		
	Unaided PTA4 (dB HL)	Aided PTA4 (dB HL)	FG (dB)	Mean F/U (Months)	Aided PTA4 (dB HL)	FG (dB)	Mean F/U (Months)
**Mean**	90.13	35.75	54.38	1.30	31.13	59.00	2.18
**SD**	3.08	7.03	6.91	0.24	7.30	7.67	1.31
**Median**	90.00	35.00	53.75	1.50	30.00	60.00	2.00
**Min**	78.75	21.25	33.75	1.00	18.75	33.75	0.75
**Max**	102.50	57.50	72.50	1.50	56.25	80.00	7.00

PTA4 (Pure Tone Average at 0.5, 1, 2, 4 kHz); F/U (Follow-up); SD (standard deviation); Min (minimum); Max (maximum).

**Table 2 jpm-12-01326-t002:** Audio Processor Satisfaction Questionnaire (APSQ).

	APSQ Dimensions					Daily Wearing Time		
	Wearing Comfort	Social Life	Usability	Total Score	APSQ after Surgery (y)	Hours	Total *n*	Percent (%)
**Mean**	8.29	9.38	9.98	9.21	2.33	**>12 h**	50	76.92
**SD**	1.25	0.84	3.40	1.35	1.78	**9–12 h**	12	18.46
**Median**	8.45	9.75	10.00	9.39	2.00	**6–9 h**	2	3.08
**Min**	4.75	6.67	6.20	6.80	0.50	**3 h**	1	1.54
**Max**	10.00	10.00	34.50	17.15	7.00	**TOTAL**	**65**	**100**

SD (standard deviation); Min (minimum); Max (maximum); *n* (number); y (years).

**Table 3 jpm-12-01326-t003:** Subjects with a complication presented in detail.

	# Subjects	# Ears	F/U Incident	Kind of Complicaton	Treatment	Time till Solved (Days)	Latest F/U no Revision (Months)
NO Comp.		47	-	No Complications	-	-	6–85 Months
	Mean ± SD						25.45 ± 29.09
	# Subjects	# Ears	F/U Incident	Kind of Complication	Treatment	Time till Solved (Days)	Latest F/U no Revision (Months)
Minor complication	01a/RONDO2	1	2.5	(A) skin pressure under magnet of AP	AP magnet change	0	21
	05b/Sonnet	2	2.1	(A) skin pressure under magnet of AP	AP magnet change	0	18
	06b/RONDO2	3	7.9	(A) skin pressure under magnet of AP	AP magnet change	0	26
	10b/RONDO2	4	8.2	(B) redness behind the auricle. otitis media acuta	antibiotics	7	35
	11b/Opus2	5	0.3	(B) retroauricular steamy skin	local desinfection	3	26
	12a/RONDO2	6	0.4	(B) redness in the upper part of wound	antibiotics, local disinfection	3	46
	12b/RONDO2	7	18.5	(B) otitis media acuta l.sin.—spont. perforation	antibiotics, ear lavage	10	46
	16a/RONDO2	8	2.7	(A) skin pressure/redness under magnet of AP	AP magnet change	0	19
	16b/RONDO2	9	6	(A) breaking the AP (fall to the ground)	replacement AP	0	19
	17b/RONDO3	10	9.8	(B) otitis media acuta catarrhalis l.dx.	local antibiotics	3	29
	24a/Opus2	11	20.9	(A) break of the AP	replacement	0	23
	26a/Sonnet	12	36.4	(B) otitis media on the right side	antibiotics	7	90
	26a/Sonnet	12	40.5	(B) small dehiscence, bleeding wound	antibiotics, local disinfection	1	90
	29/RONDO2	13	33.9	(B) retroauricular flegmona	antibiotics	10	83
	32/RONDO2	14	1.7	(A) skin pressure under magnet of AP	AP magnet change	20	26
**Mean ± SD**			**12.79 ± 13.48**		**4.60 ± 6.04**		**39.80 ± 25.39**
	**# Subjects**	**# Ears**	**F/U Incident**	**Kind of Complication**	**Treatment**	**Time till Solved (Days)**	**Latest F/U no Revision (Months)**
Major comp. n	01b/RONDO2	1	0.2	(C) wound dehiscence	resuturing	6	21
	25	2	40	(C) electrode short-circuit due to the head injury	reimplantation	7	24
	26b/Sonnet	3	2	(C) electrode dysfunction on the left side	reimplantation	120	90
	**Mean ± SD**		**14.09 ± 18.35**		**42.33 ± 54.96**		**45.00 ± 31.84**

(A) Minor: external component complication; (B) Minor: medical related complication; (C) Major: complication requiring surgical attention; F/U (follow-up); SD (standard deviation). (#) number of.

## Data Availability

All data are available in the corresponding author.

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
