# Peer review of "Long-Term Device Satisfaction and Safety after Cochlear Implantation in Children"

_jpm, 2022, doi:10.3390/jpm12081326_

Round 1
Reviewer 1 Report
The work “Long-term Device Satisfaction and Safety after Cochlear Implantation in Children” introduces the topic in a clear and accessible way - explains the issue, contains rich literature, specifies goals. While the positive impact of the daily use of a sound processor has been discussed earlier, the user satisfaction rating with APSQ is new (especially by processor type). The topic is presented very accessible, factually correct, clearly. The conclusions drawn may be applicable in clinical practice. This applies especially to the selection of a single module or BTE processor and the selection of the magnet strength in single-module processors (such assessment has not been made in the literature). It also informs both adult patients and parents of children about how long it takes to obtain adequate results. I would devote a little more attention to the issue of using this time for rehabilitation and auditory training. It should be noted that not only rehabilitation, but also audio processor (AP) fitting is a process and takes time. Literature often assumes that once hearing technology is fit, it is being worn full-time.
Specific comments:
-
- Lines 62-66: the use of APSQ in children has been recently reported in the literature (Obrycka et al., 2022)
- In the "material and methods" section and the "satisfaction-related questionnaire" subsection, please add a reference for APSQ (Billinger-Finke M. et al., 2020)
- There is no information how the questionnaire was administered and who filled the questionnaire (children/parent)
- It would be beneficial to add data logging information of usage time if available
- Figure 2: description of the horizontal axis – “functional gain” instead of “funktional gain”
- I suggest to compare your study results to the one published. Obrycka et al. is the only one existing in the literature study on the pediatric users satisfaction with the audio processor (AP) suitable for the reference results
Author Response
Dear Reviewer, thanks a lot for your valuable comments.
The work “Long-term Device Satisfaction and Safety after Cochlear Implantation in Children” introduces the topic in a clear and accessible way - explains the issue, contains rich literature, specifies goals. While the positive impact of the daily use of a sound processor has been discussed earlier, the user satisfaction rating with APSQ is new (especially by processor type). The topic is presented very accessible, factually correct, clearly. The conclusions drawn may be applicable in clinical practice. This applies especially to the selection of a single module or BTE processor and the selection of the magnet strength in single-module processors (such assessment has not been made in the literature). It also informs both adult patients and parents of children about how long it takes to obtain adequate results. I would devote a little more attention to the issue of using this time for rehabilitation and auditory training. It should be noted that not only rehabilitation, but also audio processor (AP) fitting is a process and takes time. Literature often assumes that once hearing technology is fit, it is being worn full-time.
Comment was added to the discussion, please see lines 407 – 411
Specific comments:
-
- Lines 62-66: the use of APSQ in children has been recently reported in the literature (Obrycka et al., 2022)
Citation was added (lines 66 - 68)
- In the "material and methods" section and the "satisfaction-related questionnaire" subsection, please add a reference for APSQ (Billinger-Finke M. et al., 2020)
Citation was added (line 93)
- There is no information how the questionnaire was administered and who filled the questionnaire (children/parent)
Information was added (line 104 – 104)
- It would be beneficial to add data logging information of usage time if available
Unfortunately, the majority of used AP’s (Rondo2, Opus2) does not have data logging and we therefore decided already in the study protocol to evaluate wearing time via questionnaire
- Figure 2: description of the horizontal axis – “functional gain” instead of “funktional gain”
Figure 2 was changed accordingly
- I suggest to compare your study results to the one published. Obrycka et al. is the only one existing in the literature study on the pediatric users satisfaction with the audio processor (AP) suitable for the reference results
Study results were compared – please see lines 289 - 294

Reviewer 2 Report
The study is very well founded and the results achieved with careful methodology. Although there are no doubts about the effectiveness of the cochlear implant, especially in children, there is never too much evidence about the factors that influence these good results. The conclusion reached by this study is very important for the families of implanted children; When a child undergoes this surgery, it is essential to use the device as it should be and use it daily and during wakefulness; always taking into account a period of habituation, between 3 and 6 months, in which the children will expand the daily use of the cochlear implant. From my experience with post-implantation language development in children, those who have used this device daily and during the hours that they are receiving information have achieved better results. I believe that the authors should add information in the conclusions such as: -The great variability observed among the implanted children, age of the intervention, type of device, time and model of language therapy received after receiving that implant. -the fatigue involved in processing sound in an unnatural way through a device that modifies the hearing process and that should make us think about the need for certain breaks. In fact this fatigue has not received attention from medical specialists and speech therapists. -Keep in mind and mention to the children that complying with the daily use of the device do not obtain a complete development of the oral language.
Regarding the figures, it would be important to give greater format and clarity to the following figures:
Figure 1. Soundfield audiometry. (Greater definition and size).
Figure 2. Scatter plot correlation functional gain (FG) vs daily device use. (Larger size).
Author Response
Dear Reviewer, thanks a lot for your valuable comments.
The study is very well founded and the results achieved with careful methodology. Although there are no doubts about the effectiveness of the cochlear implant, especially in children, there is never too much evidence about the factors that influence these good results. The conclusion reached by this study is very important for the families of implanted children; When a child undergoes this surgery, it is essential to use the device as it should be and use it daily and during wakefulness; always taking into account a period of habituation, between 3 and 6 months, in which the children will expand the daily use of the cochlear implant. From my experience with post-implantation language development in children, those who have used this device daily and during the hours that they are receiving information have achieved better results. I believe that the authors should add information in the conclusions such as: -
The great variability observed among the implanted children, age of the intervention, type of device, time and model of language therapy received after receiving that implant.
Information was added, please see lines 332 -335
-the fatigue involved in processing sound in an unnatural way through a device that modifies the hearing process and that should make us think about the need for certain breaks. In fact this fatigue has not received attention from medical specialists and speech therapists. -Keep in mind and mention to the children that complying with the daily use of the device do not obtain a complete development of the oral language.
Discussion was extended accordingly, please see lines 358 – 361
Regarding the figures, it would be important to give greater format and clarity to the following figures:
Figure 1. Soundfield audiometry. (Greater definition and size).
Figure 1 was changed accordingly (figure size and text size was increased)
Figure 2. Scatter plot correlation functional gain (FG) vs daily device use. (Larger size).
Figure 2 size was increased
